# Trauma Responses in Social Choreography: Accessing Agency and Opportunities for Healing through Mindful Embodiment

## Catherine Cabeen

Department of Dance, Marymount Manhattan College, New York, NY 10021, USA; ccabeen@mmm.edu

**Abstract:** This article responds to the questions: how does trauma that is long-held in the body affect social choreography? And how can awareness of this intersection guide us towards individual and collective healing practices? Embodied trauma responses, commonly referred to as fight, flight, freeze, and dissociation, initially function as potentially lifesaving responses to external threats but all too often become engrained in how people move through the world and relate to one another. When these patterns of engagement become habituated, they affect the improvisational scores inherent to social choreography. Exploring trauma responses through the lens of social choreography invites increased awareness of how these patterns of behavior affect our relationships and communities. Through this awareness, the possibility of agency is increased. This essay continues the work of somatic and cultural scholars Resmaa Menakem, Staci K. Haines, and Zhiwa Woodbury, among others, whose research points to multiple continuums between how trauma is individually embodied and cultural dynamics we are experiencing globally. Drawing on the somatic work of Peter Levine and Bessel Van der Kolk, whose theories have revolutionized trauma healing, this essay offers accessible pathways to trauma sensitivity that readers can experiment with to consciously refine their own roles in social choreographies ranging from interpersonal to cultural interactions.

**Keywords:** social choreography; trauma; somatic practices; mindfulness; healing

## 1. Introduction

In January of 2023, I was invited to Puerto Rico by NYU's Faculty Resource Network to teach a week-long workshop to fellow educators entitled Trauma-Informed Pedagogy (TIP). The workshop drew upon my experience as an educator and somatic movement therapist, in contexts that include public, private, and prison education systems. When I was invited to write for this Special Issue on *Choreographing Society*, immersed as I was in research that supported the TIP workshop, I could not help but think about the ways that trauma lingers in people's tissues, unconsciously affecting their individual behavior, which in turn informs interpersonal and transpersonal dynamics. As such, trauma can be understood as a force that "choreographs" our social interactions. Choreographer Bill T. Jones, who I have worked with in varying capacities for 25 years, defines choreography as "the movement of people and things in time and space". This spacious definition invites us to analyze intimate relationships and social structures, as well as the movements of dancers on concert stages, through the lens of choreography. The performance studies lens of Social Choreography more specifically examines the underlying patterns in social interactions and engages choreographic terminology in order to invite creative alternatives to surface via this analysis (Kliën 2017). This writing explores the ways in which trauma so often forms and informs underlying patterns of social interaction and invites the reader to imagine new potentialities in relational dynamics that might be available by healing and/or tending to the traumas we carry.

The TIP workshop was designed to help educators better care for students, who we all realize are navigating an increasingly traumatizing world, evidenced by school shootings, climate change, political polarization, and a pandemic, among other factors.

However, as we began to collectively investigate the ways in which traumatic retentions inform behavior, the community of learners was struck by the essential realization that we all needed to acknowledge the histories of trauma in our own bodies before we could better support the students we work with. Because trauma tends to affect behavior in ways that are unconscious and/or habituated, the first step in trauma healing it is to see traumatic retentions clearly for what they are and to recognize what activates them. In the TIP workshop, in addition to discussing the science, definitions, politics, and cultural contexts of trauma, we engaged in mindfulness and somatic practices to uncover how trauma informs our own participation in various social choreographies. The workshop explored how trauma can underlie implicit biases and employed somatic practices in the framework of how they can support social justice work. This essay outlines many of the trauma-related topics we explored in the TIP workshop and introduces mindfulness and somatic practices as a foundational step in the multi-faceted work of trauma healing. This essay responds to two questions: how does trauma that is long-held in the body affect social choreography, which refers to intimate interactions, group dynamics, and collective cultural expressions? And how can awareness of this intersection guide us towards individual and collective healing practices? I argue that embodied trauma responses are enacted in social choreographies in ways that are most often unconscious, but increased agency in social choreographies becomes possible when we recognize and tend to traumatic retentions in our bodies and communities of practice.

## 2. Definitions: Trauma, Social Choreography, and Somatic Mindfulness

Trauma can be defined as any experience, or series of experiences, that overwhelms and injures the nervous system of a sentient being. The nervous system injuries caused by trauma are held in the body, stimulating embodied trauma responses such as flashbacks, panic attacks, and/or dissociation, which dramatically or subtly affect behavior, interactions, and self-perception. "When something happens to the body that is too much, too fast, or too soon, it overwhelms the body and can create trauma" (Menakem 2017, p. 7). This broad definition of what causes trauma, offered by social worker, author, and activist Resmaa Menakem, embraces the wide range of experiences that can cause trauma, not to conflate them, but to invite the reader to consider the myriad ways that interactions with our environments can overwhelm our bodies. Embodied trauma responses can result in Post-Traumatic Stress Disorder (PTSD), which is recognized by the American Psychiatric Association in its Diagnostic and Statistical Manual of Mental Disorders as an intense, individual mental and physiological condition (American Psychiatric Association 2013, pp. 271–72). However, I primarily use the terms embodied trauma responses or traumatic retentions in this writing, rather than PTSD, to both recognize the spectrums of intensity of traumatic retention and to honor the interdisciplinary work of clinical psychologists, social workers, somatic therapists, and cultural theorists who articulate trauma as having interpersonal, generational, and cultural manifestations. Embodied trauma responses are not contagious, but as they are enacted, they can deeply impact relationships and communities. As trauma moves between bodies through these interactions, it becomes a player in social choreographies ranging from intimate interpersonal duets to the dances of power inherent to systemic oppression.

Social choreography is a framework of cultural analysis that invites us to explore questions of agency, power, and habituation within social patterns of interaction. For example, who/what is the choreographer of social choreography? In other words, who/what chooses, controls, or plans the frameworks and/or dynamics of social interactions? Who are the dancers or participants that are interacting? What leeway for improvisation and/or agency exists within various social choreographies for these dancer/participants? How long has a given choreography been being rehearsed? To what end? And perhaps most importantly, once we see the structure and impact of a given social choreography, what other options might we have? Embodied trauma responses are often so foundational to such well-rehearsed social choreographies that they are imperceptible to the individuals

and communities enacting them. As a result, social choreographies can unintentionally get caught in loops that reinforce traumatic retention. However, using social choreography as a cultural studies lens has the dual purposes of seeing what is and envisioning what else is possible. Hence, I bring this framework of social choreography to embodied trauma responses to empower us to both recognize the ways in which trauma choreographs our interactions and, through this recognition, to inspire us to imagine other ways we might engage with one another. I am undoubtedly interested in the question of what creates agency within choreographic structures because my own background includes 30 years as a professional dancer. Throughout my professional practice, I was involved in choreographies in which I was tasked with the exact replication of pre-determined steps and other choreographies in which I had the opportunity to make choices. I was involved in hierarchical creative practices and collaborative ones, and I distinctly preferred the latter. Having participated in concert stage choreography and creative processes that seek to embody sociopolitical forces, analyzing social interactions through a choreographic lens comes as a logical extension of my practice as an artist. Throughout this essay, I refer to people as dancers and to social engagement as a dance. In the same way that Social Choreography uses choreographic terminology as both a metaphoric and literal description of dynamic relationships, I assert that we are all dancers in the dance of life.

A growing number of researchers and therapists are finding that mindfulness and somatic awareness can help us to bring our habituated patterns of posture, speech, and engagement to light. These embodied practices can serve as potent first steps in healing the traumas our bodies carry and empowering more conscious movement through the social choreographies of our communities. While trauma often cannot be undone, embodied trauma responses can be tended to. By first recognizing where they surface and then working with these trauma responses via mindful somatic practices that support embodied agency in how we respond to various stimuli, we can shift from being danced by our trauma responses to dancing with them. Often this deep, personal work happens in a one-on-one context with a somatic movement therapist or somatic psychologist. Intentional group settings can also be effective environments to engage with somatic healing. Somatic methodologies often employ movement, touch, mindfulness, vocalization, and discussion. This essay is intended as an overview of why somatic, body-based therapeutic modalities are valuable in trauma healing and as a literature review that cites numerous resources through which interested readers can dig deeper into a wide array of somatic approaches used in trauma healing. As we explore trauma as something that affects individuals, families, and communities, it is worth clarifying that somatic practices are one part of a healing practice and that they are intended to support individuals and empower further action. They are not an end in and of themselves, but a prerequisite for effective engaging with challenging social dynamics. Mindful somatic practices support intentional action in the world. Social change requires complex community organization and sociopolitical engagement. Somatic practices help support individuals that are involved in that work by stabilizing their nervous systems so that they can have the stamina to engage with the vast work of cultural innovation.

Vicarious or secondary trauma, the kind often experienced by therapists, first responders, and nurses who witness trauma happening to others, can be as detrimental to the nervous system as a first-hand experience of trauma (Treleaven 2018). Even discussions of trauma, not to mention the graphic images that populate many of our social media feeds these days, can activate our own trauma responses. So, as we journey into this subject matter, it is important to define and briefly practice "Resourcing". This somatic practice invites us to identify anchors for our attention: objects, songs, mantras, breath, smells, supportive physical sensations, nature, the face of someone you love, etc., and to practice bringing our attention to that supportive "Resource". Our ability to do this is key to our own emotional self-regulation. When we are unable to self-regulate, we can dramatically affect the social choreographies around us by spinning out in our own reactive dances. Additionally, we cannot help others through their challenges when we ourselves

are activated. Like all somatic practices, Resourcing is just that: a practice. Our ability to direct our attention grows stronger over time if we practice.

> *I invite you to pause your reading for a moment to create a short list, in your mind or on paper, of Resources in your life that you can return your attention to anytime you start to feel uncomfortably activated. Take a few breaths, centering your attention on one of your Resources.*

## 3. The Space between Stimulus and Response

Mindfulness and somatic practices are vast disciplines with many overlaps, which are regularly engaged as a part of healing trauma and Post-Traumatic Stress Disorder (PTSD). Mindfulness can be defined as kind attention to the present moment. The founder of Mindfulness-Based Stress Reduction, Jon Kabat-Zinn, defines mindfulness more specifically as "awareness that arises through paying attention, on purpose, in the present moment, non-judgmentally" (Kabat-Zinn 2013, p. 17). Mindfulness is a secular tool that emerged from Eastern religious practices as they intersected with the West. Somatic practices often utilize mindful attention directed specifically towards the physical body as it interacts with its environment. Somatic work is distinguished from other forms of exercise through its engagement with the body as a vital, sensitive being rather than as a mute object that needs to be fixed and/or maintained.

> *After reading these instructions, I invite you to pause your reading and take a few deep breaths. Notice what you are feeling in your body right now. This is often referred to as "Open Awareness", just allowing whatever arises to be noticed and named. After you name a sensation, see if you can let it go to make space in your consciousness for other sensations. You may notice tension or weight, you may feel the texture of your clothes or your seat; you might hear or smell things in the environment around you; you might feel the movement of your breath in your body, or even your heartbeat. Try to notice and name these sensations without attaching a story to them; try not to label them as good or bad, but just as what is: crossed legs, tension in the neck, traffic outside, fullness or hunger in your stomach, warmth or coolness in the air of the room... Numbness or a lack of sensation may also be what you notice. There is no right or wrong here, just a moment to check in.*

As you give yourself a moment to feel the sensations of the body, you may notice a tendency to judge what you feel. We tend to move towards pleasure and away from pain, but mindfulness and somatic practices instead invite us to register all these sensations with equanimity. The goal is to acknowledge feelings without getting lost in the stories we have created, and trained ourselves to believe, about what it is we are feeling. This practice creates a tiny space between our felt reaction to something and our enacted response to it. The ability to pause in this space is a prerequisite to our ability to disempower trauma responses, among other conditioned reactions. In attending to this tiny space, we build our capacity to make choices within the social choreographies of our lives.

Trauma often causes dissociation from the body. Initially, this is a protective reaction that enables us to survive overwhelming experiences, but if an individual gets stuck in dissociation, it can lead to anxiety, fatigue, and a lack of engagement with life. Mindful somatic practices can invite dissociated individuals back into communion with their bodies. Injury, illness, and other life challenges are experienced through the body, *and* our body is the instrument through which we experience pleasure, joy, and connection, so opening to our awareness of embodied sensation increases our access to the delights of life as well as to the challenges. Social choreographies are also experienced through our bodies, and the more awareness we bring to our positionality within the dance, the more we can make choices about how we participate in its unfolding. Maintaining an awareness of our bodily sensations in the present is challenging for most people as the legacy of Cartesian Duality, with its hierarchical separation of the mind over the body, is a strong organizing principle in Western culture and institutions. In addition, the social forces of capitalism and technology

amplify disembodiment in contemporary culture. For people who are struggling with PTSD, awareness of embodiment is particularly tenuous, but it is simultaneously the key to significant healing.

Peter Levine and Bessel Van der Kolk are the authors of seminal texts that forged a path for our contemporary understanding of the connection between trauma healing and embodiment: *Waking the Tiger* (Levine 1997) and *The Body Keeps the Score* (Van der Kolk 2014), respectively. To appreciate their methodologies and breakthroughs, it is helpful to have a basic understanding of how traumatic experiences affect our body–minds. The role of the body in traumatic retention, and, therefore, in trauma healing, cannot be over emphasized as trauma responses are not controlled by our rational minds. Overwhelming experiences activate the limbic system of our brains. Its functions are instinctive and operate before our prefrontal cortex's rational cognition can register what is happening. The limbic system is hard-wired to our organs via the Vagus nerve. Over 90% of the Vagus nerve's fibers are afferent, meaning that they relay information from our organs to our brains (Levine 2010, p. 141). So, it is our "gut instincts" that inform our brain when we feel threatened. When our bodies sense we are in danger, the Vagus nerve activates our sympathetic nervous systems (SNS) in what are commonly known as the fight, flight, or freeze responses. These responses are incredible and effective evolutionary adaptations to direct threats. However, they are meant to support a body in the short-term and then switch off. Embodied trauma responses form when these stress responses are chronically activated, resulting in a traumatic retention.

In her book, *The Politics of Trauma*, Staci K. Haines described how our 'soma', a combination of our living body and its actions, can be shaped by this chronic activation of our stress responses. Over time, this can result in hypervigilance, or chronic dissociation that feels "normal" (Haines 2019, p. 97). The body caught in a fight response has a forward-leaning posture, unrelenting muscular tension, and is only interested in combative social choreographies. The body caught in a flight response leans away from contact, ends relationships and engagements inexplicably abruptly, and only participates in social choreographies that require little commitment and have multiple opportunities to exit. A body caught in a freeze response may experience considerable numbness, regularly find their mind going "blank", and will avoid social choreographies that are overly stimulating (Haines 2019, pp. 100–13). These engrained trauma responses can have high costs, ranging from fatigue, to anxiety, to social isolation. Haines stated, "Traumatic experiences are always happening within a social context and shaped by social conditions. They are nearly always perpetuating the "rules of engagement" of our social conditions (Haines 2019, p. 95). In other words, trauma in our somas can take over the choreographer's role in any social choreography, directing the posture, agency, and quality of interaction of the dancers. Stress responses can become chronically activated and take hold of our somas due to several factors, which I explain further below as being due to miscommunication between the parts of the brain that help us to timestamp our experiences, a survival instinct being initiated biochemically but then thwarted, or a threat being persistent and unrelenting. In each of these situations, focused, embodied practices can help to re-regulate the nervous system.

## 4. The Body Is Present

Dr. Noel Larson stated, "If something is hysterical, then it is usually historical" (Menakem 2017, p. 42). When we find our social choreographies to be infused with either much more energy or much less energy than makes sense for the situation, we are often dealing with a trauma response that is fueled by a miscommunication between two parts of the limbic system: the amygdala and the hippocampus, which usually function together to regulate shifts in and out of stress responses. The amygdala can be compared to a smoke detector, sounding an alarm when it perceives a threat. The amygdala also records general features of the experience, like a blurry photograph. The hippocampus recalls many more details of the event and, importantly, provides a time stamp on the event. In the case of a traumatic event, the communication between these two parts of the

brain is disrupted. As a result, a generalized memory of something being a threat may not receive a timestamp, meaning that we do not have a firm grasp on it being a memory. For example, if you were listening to a favorite song when you were suddenly threatened, it could develop that rather than having a complete memory about the event, any time you hear that song, your body–mind could perceive a threat and initiate a fight, flight, or freeze response in the present. This can manifest as the kind of panic attack often associated with PTSD or a shrinking and pulling away from vital engagement with the present moment.

Van der Kolk's work centers on the traumatized body's inability to distinguish past from present. He found, through his extensive therapeutic practice with veterans, that the body is a key gateway to the present moment. By inviting veterans who were struggling with PTSD to focus on physical/embodied sensations in the present, Van der Kolk was able to help many people who were suffering with PTSD to develop skills to stay grounded in the now. This approach to embodied mindfulness harnesses the power of that tiny space between stimulus and response to empower the survivors to recognize the trigger and trauma response but not get carried away with it. Instead, with practice, they developed the agency to self-regulate in the face of the traumatic activation as gradually, their somas were able to correctly categorize the trauma they experienced as part of their past, distinct from a "safe" present.

*A useful entry point into feeling one's body in the present is found through a somatic practice called Grounding. I invite you to pause after reading these instructions and feel the parts of the body that are foundational to your posture (feet, thighs/pelvis if one is seated). Allow your awareness of this physical foundation in contact with the floor or chair to amplify your connection to the earth. Feel yourself being supported by the ground beneath you.*

## 5. The Body's Need to Move

In addition to a confusion around the past feeling present, we can also find ourselves caught in trauma responses due to survival instincts being initiated by the nervous system but then thwarted. When we need to fight or escape, and we successfully do so, the body can move through the intensity of arousal and then process the experience. Many animals face life or death situations regularly in the wild but do not end up with PTSD because they engage their bodies fully in their struggle for survival and then, once safe, literally shake it off (Haines 2016, p. 20). However, humans are often not able to run or fight or shake. For example, if a traumatic event happens to a child, where they cannot escape the situation and are being silenced, they are likely to end up with PTSD. In addition, if something happens to us as adults and due to social mores and behavioral expectations, we repress our body's need to fight or escape or shake, we are also prone to develop PTSD.

Here we find that social choreographies can be problematic or liberating, all depending on what dances we participate in. The social choreographies that uphold white supremacy, toxic work environments, and abusive relationships can cause us to repress our emotional responses, leading to PTSD, while other social choreographies of protest and solidarity can help us to express our need to fight, to change our situation, or even to just "shake it off". Dr. David Berceli is a leading expert on trauma recovery and the creator of Trauma Release Exercises (TRE). These movement practices engage repetitive movements in big muscle groups of the body, bringing them to fatigue, which causes them to shake and, as a result, potentially release long-held trauma (Berceli 2008). Somatic practitioners often encourage a more literal approach to shaking things off and regularly invite an opportunity to manually shake out the body as a form of bodywork or to engage in a shake-based improvisational dance.

*I invite you to stay seated or stand, but pause your reading for just a moment and… shake it out! Shake your hands, your head, shimmy your chest or hips, shake one part of the body or several all at once. You might notice sounds arise from the shake, voice them as a part of the release.*

Grounding and restorative shaking to complete a survival response are also essential to the work of Peter Levine, PhD. Levine developed his therapeutic methodology, Somatic Experiencing, on the foundation of psychologist and neuroscientist Steve Porges' Polyvagal theory, which explores the dual functions of the Vagus nerve (Levine 1997). As noted above, the Vagus nerve connects the brain to the organs of the body. This nerve, the longest in the body, has two branches; the Dorsal Vagus which activates the SNS (sympathetic nervous system, fight/flight) and the Ventral Vagus which corresponds to the PNS (parasympathetic nervous system, rest/digest). When the body is caught in a trauma response, it has limited access to its social engagement systems, which are animated by the Ventral Vagus/PNS. Much of Levine's work is dedicated to developing methodologies to help a body shift into its Ventral Vagal state, downregulating the SNS. Importantly, for the connection between trauma healing and social choreography, Levine found that when two people interact, if one can maintain a settled nervous system or Ventral Vagal state, the two systems can coregulate. He wrote of this experience, "...the support of a calm present other allowed the powerful and profoundly restorative involuntary reactions to emerge and complete themselves" (Levine 2010, p. 13). Re-activating the social engagement system after it has shut down due to a traumatic experience or the triggering of a traumatic retention is virtually impossible to do alone. Intimate social choreographies that invite eye contact and/or gentle, consensual physical touch, and which are initiated by a player(s) with a settled nervous system, can foster calm states that move between bodies and, eventually, potentially harmonize a community. However, this is also true of an activated system. "When one settled body encounters another, this can create a deeper settling of both bodies. But when one unsettled body encounters another, the unsettledness tends to compound in both bodies. In large groups, this compounding effect can turn a peaceful crowd into an angry mob" (Menakem 2017, p. 39). Trauma's ability to move between bodies is central to its ability to affect social choreographies.

Overwhelming events are most often the cause of trauma; however, what is traumatic to one person may barely affect another. This is due to biological and behavioral factors, both of which affect the individual's role in the social choreographies of their community and culture. The field of epigenetics is contributing greatly to our current understanding of generational trauma. This research explores changes in an organism's gene expression and has illuminated dramatic realities in recent decades about how traumatic retentions can be encoded in our DNA and passed on from parents to children for generations (Yehuda 2016). Each of us individually and the larger cultures we live in have distinct tolerances and sensitivities that can cause a given situation to leave a traumatic imprint on one person or group and not on another. The unique combination of stimulants that might activate a trauma response in each of us leads to the complexity of social choreography involving trauma.

## 6. Social Choreography as Traumatic Retention

Social choreographies are often intentional and well-rehearsed, such as in the case of political pageantry or religious rituals. Social choreography can also include structured improvisations, wherein the structure, or score, is the cultural conditioning that directs the agency of the players.[1] We see cultural conditioning emerge as this kind of choreographic score in intimate relationships where histories of gendered power may be consciously or unconsciously embodied and perpetuated and in institutions where historic hierarchies of gender, race, ability, and class are regularly reinforced. Depending on the intersectional body politics of the players, the scores for these social choreographies can be either empowering or traumatizing. In addition, these scores may have emerged from trauma.

"Unhealed trauma acts like a rock thrown into a pond; it causes ripples that move outward, affecting many other bodies over time. After months or years, unhealed trauma can appear to become part of a person's personality. Over even longer periods of time, as it

is passed on and gets compounded through other bodies in a household, it can become a family norm. And if it gets transmitted and compounded through multiple families and generations, it can start to look like culture. But it isn't culture. It's a traumatic retention that has lost its context over time" (Menakem 2017, p. 39).

With this statement from *My Grandmother's Hands*, Resmaa Menakem invites us to look closely at the aspects of ourselves, our families, and our cultures that we take as fixed and instead explore them as choreographies that can be transformed. Trauma creates, and is created by, embodied conditioning that is deeply intertwined with, and inherent to, the dynamics of social choreographies. These play out between bodies in families and then "ripple out" to be constitutive of larger communities and institutions.

When we recognize social practices as being engaged in choreography that we have the potential to co-create, rather that replicate, we open our social and cultural practices to possibilities of change. Scholar Michael Kliën stated: "The practice of Social Choreography deals with the uncovering of underlying social relations and patterns—the choreography of the social—through embodied practices, and always, engages these dynamics for new social choreographies to emerge simultaneously" (Kliën 2017). The potential that Kliën observes between recognizing underlying patterns and opening to the possibility for new patterns to emerge resonates with the process of healing trauma, wherein embodied trauma responses must be brought to consciousness to be transformed. In my experience of teaching about trauma, most people are entirely unaware of how much of the choreography of their lives is directed by it.

In my own experience, recognizing that I habitually reacted to family "fight" dynamics with a "flight" response has been key to making my family dynamics healthier. Over time, recognizing the choreography of how these traumatic retentions feed one another empowered me to disrupt them in interpersonal settings, creating a conscious shift in my family dynamics that supports all parties involved. I also fall into this "flight" reaction in response to homophobia and misogyny in the workplace. Here, my individual trauma interacts with deeply engrained social patterns of oppression: "traumatic retention that has lost its context over time" (Menakem 2017, p. 39). In this larger social context, recognizing the pattern is not enough to disrupt it. Collective efforts such as unionization, enacting legal protections, or developing social safety nets that would empower individuals or groups to leave toxic environments are needed. Menakem argued that social choreographies of racism, hierarchy, gendered power, and heteronormativity all stem from decontextualized trauma. While it is true that "hurt people, hurt people", many of the oppressive social choreographies that have been created by "hurt people" are so deeply engrained in contemporary society that they take immense amounts of organized social pressure to shift. While being grounded and Resourced can help an individual and community to have the clear vision and endurance they need for this kind of social activism, it bears saying that while the work of activism is supported by somatic mindful practices, social organization certainly requires additional strategic labor.

Author Ronald Purser takes this one step further in his book, *McMindfulness*. Here, he argues that mindfulness and somatic practices have been coopted by capitalism to make individuals think that their unhappiness is a personal failing, rather than a response to systemic injustice. (Purser 2019) He suggests that practices like Resourcing can be used to make an individual comfortable *enough* to tolerate injustice and abuse. If we can be grounded in the face of trauma, Purser argued, why would we ever address the traumatizing social systems of oppression when we can instead just keep working on our ability to ground ourselves? Purser made valuable observations about the danger of conflating a tool with a solution. The goal of somatic, mindful practices is not the sense of grounding that they produce, but what we can *do* in that grounded state to affect our social choreographies.

## 7. Trauma Terminology and Its Cultural Evolution

The evolution of the language we use to describe trauma, and its shifting classification as a medical diagnosis, affect our ability to make sense of overwhelming experiences and also aid individuals and communities in seeking treatment and support. The *Diagnostic and Statistical Manual of Mental Disorders* (DSM) was first published in 1952. The first edition, the DSM-1, did not include any mention of trauma or PTSD, but it did mention a "Gross Stress Reaction" that can occur in response to severe physical demands or extreme emotional stress such as in combat or in civilian catastrophe (fire, earthquake, explosion, etc.) (Finch 2017). This mention of combat functioned as an implicit critique; war is inherently psychologically harmful and is likely to cause a 'gross stress reaction' in any person that undergoes such an experience. Not surprisingly, the diagnosis disappeared from the DSM-II of 1968, published in the heat of the Vietnam War protests. Post-Traumatic Stress Disorder was then first included in the DSM-III in 1980. This was the result of heavy lobbying by veterans and veteran associations, who needed the diagnosis to ensure treatment by the VA. This lobbying to recognize the effects of traumatic experiences was supported by the second wave of feminism, which began to push conversations about rape and childhood sexual abuse into the public eye in the 1970s and 1980s (Treleaven 2018, p. 55). As a diagnostic criterion was established, therapists worked with veterans and survivors of various kinds of abuse to refine the terminology and definition of PTSD in the DSM.

The struggle for the recognition and understanding of trauma and its myriad/complex causes continues through volatile social choreographies of protest, as well as in intimate, immersive social choreographies of care. While the DSM-III definition of PTSD was limited, it laid a foundation for research into the care and treatment of trauma that continues to evolve. The *DSM-5*, the 2013 update to the DSM, defines trauma as, "exposure to actual or threatened death, serious injury, or sexual violation" (American Psychiatric Association 2013). In recent years, the intersections of trauma research and anti-oppression work have built upon one another. While a singular event can leave an individual traumatized, trauma is now understood as something that can also emerge from experiences that happen over time. Monnica Williams, PhD, has written extensively on the fact that microaggressions, subtle widespread acts of discrimination, have a compounding effect that can result in traumatic symptoms (Williams 2021). Race, class, gender, sexuality, and ability are all intersectional factors that put individuals and communities at risk for potentially traumatic generational and systemic discrimination. Even in a well-intentioned healing space, the body politics of therapists and clients and the inherent power dynamics between them can activate trauma responses. Martha Eddy and Carol Swann began developing Social Somatics in the mid-1990s to actively address systemic oppression as a social choreography that is present between somatic therapists and clients. "Social Somatics uses awareness of cultural complexity and contexts of privilege and oppression to engage in creative and embodied action. These practices aim to bridge disconnections and transform cycles of injustice into new paradigms of mutual respect for all" (Eddy 2016, p. 234). Social Somatics practices invite a deep awareness of our positionality within social choreographies, which is an essential aspect of contemporary trauma healing.

## 8. We Are All in This Together

While Levine and Van der Kolk's work revolutionized trauma healing practices by shifting from a "talking therapy" model to a more embodied one, their approaches depend on the trauma being in the past and the present moment feeling "safe". However, as Williams has articulated, trauma that emerges from systemic/cultural conditions such as racism, homophobia, misogyny, ableism, and political polarization can also lead to PTSD. This realization in the therapeutic field has added deep complexity and an awareness of intersectional identities to trauma healing because these traumatic ideologies of oppression are both historic and also dangerously present (Tippet 2021). In addition, the growing and ongoing threat of climate change exposes all of humanity not only to the traumatic threat of climate-related catastrophic events, but also to global annihilation. "After all, when we

say the words 'climate change', are we not talking about a pervasive, continual assault on the global biosphere? One that threatens mass extinction and overwhelms our emotional capacity? Is this not the very definition of trauma?" (Woodbury 2019, p. 1). In her work on climate trauma, Zhiwa Woodbury proposed that the increasing polarization of political debate is a form of global fight/flight response and frames our inability to combat climate change as a form of global dissociation. Here, we see humans dancing numbly in social choreographies of consumption to the beat of neo-liberal capitalism, a deadly dance if there ever was one. The lens of climate trauma opens our apertures to consider PTSD at a global level. In the case of climate trauma, we may or may not be able to reverse the damage that humans have done to the earth, but we will be far more likely to co-create social choreographies that can navigate new climate realities if we first wake up from our collective dissociation.

Woodbury argues for shifting our naming of 'climate change' to 'climate trauma' because trauma is something that can be tended to. Whether we are talking about individual, generational, or cultural trauma, the path to healing begins with recognition of which behaviors are actually decontextualized trauma responses. Mindfulness and somatic practices involve many tools that can help us in this sensitive work, giving new valence to somatic practices such as Resourcing that was introduced at the beginning of this essay. Resourcing is essential in order to have a sense of relative safety in which to invite folks to be present so that further collective action can be organized and taken from a grounded place. A core practice of Trauma-Informed Pedagogy is to begin every class, workshop, or semester by defining Resourcing and having students create and share lists of their Resources. This helps communities to be built on an acknowledgement of what its members love and a shared sense of joy. From this foundation, collective action can be organized and sustained.

### 9. Conclusions: Let Us Get to Work

From a clear and grounded place, we can better understand the origins and intentions of the social choreographies we engage with. As we shift our habitual participation in them to a conscious, co-creative role, perhaps we can re-choreograph dances of oppression and hierarchy to be more inclusive. We could celebrate the social choreographies of different cultures rather than placing them in competition with one another. This potential exists, but it requires each of us to do our individual work to heal the traumas our bodies carry. To recognize traumatic retentions, one needs to slow down and direct one's attention to their embodied sensations as they shift. This is a significant ask in our "time is money" contemporary social choreography, but it is time well-spent. Menakem stated, "One of the best things each of us can do for ourselves and our descendants, is to metabolize our pain and heal our trauma. When we heal, we may spread our emotional health and healthy genes to later generations" (Menakem 2017, p. 55). In the *Trauma-Informed Pedagogy* workshop, turning toward introspection and engaging with mindfulness and somatic practices allowed the workshop participants to both recognize patterns of traumatic retention in their own bodies and invite new choreographies to emerge. This work empowers educators to be Grounded and Resourced so that their settled nervous systems can support the learning environments they facilitate. The work of healing the traumas we carry is not easy, but it is a worthy calling to empower each of us in our transformation from unconscious participation in historic dances of power and wounding to the conscious co-creation of healthy, empowering social choreography.

**Funding:** This research received no external funding.

**Data Availability Statement:** No new data were created or analyzed in this study. Data sharing is not applicable to this article.

**Conflicts of Interest:** The author declares no conflict of interest.

## Notes

[1]     In choreography, a "score" usually refers to lists of physical prompts or limitations that performers adhere to, linking the dancers' intensions while still giving them room to improvise within the assigned tasks.

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
