# Peer review of "Trauma Responses in Social Choreography: Accessing Agency and Opportunities for Healing through Mindful Embodiment"

_arts, 2024_

Round 1

Reviewer 1 Report

Comments and Suggestions for Authors

I appreciate the opportunity to review your piece. Thank you for the time and focus you put into writing it. 

Overall, I find it an important contribution to the growing literature connecting the systemic with the embodied. I find the article helps to widen the scope: how the social and structural lives in the tissues, how trauma is both personal and social, and ways we can attend, listen, be, and heal, to open choices in the social choreography.  

I appreciated you adding moments of practice through the piece.  For me it increased my pleasure as I read it, and helped me to integrate your words and ideas.

There are two areas where I'd like to invite your further thinking/ reflection.  

You powerfully named the personal and intergenerational impact of trauma and systemic oppression. You named how these become embodied and can then become "culture."  And, in the place of intervention, of healing, you left it solely at the individual and relational level. If the reader does not have an understanding of structural change, or knowledge of historical movements for equity, it could be read as if personal changes can change structural oppression. 

This is one of the areas in which, I think, we need to be careful, as we more consciously intersect trauma healing and structural oppression.  How we as people, bodies, and relationships change, is deeply connected to embodied healing. How nations, economies, and governing changes are connected to larger groups of people co-creating and mobilizing at a very different scale. Those people/ leaders need access to embodied trauma healing too, for sure.  And, trauma healing and broad scale social change require very different processes.  I think those of us writing in this field need to continue to distinguish the interventions at different scales so they don't get collapsed.

The 2nd area of reflection is about clarifying what somatic awareness and mindfulness can do, and not do, amidst other necessary aspects of embodied trauma healing.  What is the role of healing with and amidst others, and practitioners -- deep mending around relationships?  What is the role of, and container needed for, unwinding the trauma, and emotions from our bodies?  What is the role of touch? What is the role of learning embodied skills that trauma and oppression didn't teach us?  

It is not that your article needs to cover all of these in the beautiful depth in which you spoke about somatic awareness based practice, but I do think it is important to speak to the range of what’s needed in embodied healing, so readers' scope is clearer.

Thank you again, and please keep writing. 

Author Response

Thank you for this clear and supportive critique.

I very much appreciate your articulation of the difference between individual healing and collective social transformation and the danger of conflating the two. I have added clarification on this point to several sections of the essay; speaking to the importance of personal healing and the complexity of systemic change. I added a brief nod to Ronald Purser's McMindfulness, which is an explicit critique of mindfulness as a tool for social change, as a way to frame my argument (which I hope is clearer now) that somatics and mindfulness are tools that can help prepare us to take further action from a conscious and considered place, rather than Grounding/Resourcing being the end goals. 

Thank you also for pointing out that my writing neglected to offer frameworks for trauma healing work! In the workshop that inspired the writing we worked as a collective, but I stated many times that healing takes place over time and that often an individual/one-on-one therapist needs to be engaged to really unpack the depth of our traumatic retentions. I added language to this effect to the essay in order to better frame what the essay addresses and the work, beyond reading an essay, that many folks wanting to heal traumatic retentions will be called to do. 

I hope that these edits address your concerns. Thank you again for your valuable feedback.

Reviewer 2 Report

Comments and Suggestions for Authors

Comments on the Quality of English Language

Author Response

Thank you for this clear and pointed critique. 

I appreciate your articulation of the writing being for a particular, niche audience. Indeed, it was written in response to a specific call for papers from a community I am in regular dialog with, to be part of a collection of works on "Choreographing the Social." That said, I of course would like my writing to be as accessible as possible. I added more information that frames my positionality within the community of practice that requested the writing in order to be transparent about the context of the writing. I also added definitions of "choreography," "choreographer" and and explained my use of the terms "dance" and "dancer". These definitions are focused on how I am using the terms within the essay, as entire volumes have been written on the definitions of these words.

I feel it is important to define trauma broadly in this essay because the writing is aimed at highlighting the wide ranging ways traumatic retentions affect our social interactions and cultural landscape. Certainly, thoroughly unpacking all of these dynamics is beyond the scope of a single essay. I agree that my lack of experience as a academic writer is clear here in my overly ambitious scope. Thank you again for encouraging me to be more specific. I have hopefully clarified, the focus of what I am writing this essay for by being clear that mindful and somatic approaches are tools that can support action in the world, and that I do not see grounding as some kind of magical cure in and of itself.  

I have renamed and edited the section that explores the "history" of trauma coming to be a diagnosis in the DSM to better frame the points in that history I am interested in- namely how social movements affected the medical terminology.

Yes- Cartesian Duality is at the heart of this writing, but I'm fascinated by the way you read the essay as supporting dualism instead of arguing against it. It is great that you as a reader feel that it is self-evident that trauma resides in the body. However, one of the purposes of my doing this writing is that the use of embodied therapeutic modalities is still considered to be very progressive in psychotherapy. In fact, whole new fields of Psychophysiology and Somatic Psychology are currently emerging, in response to the authors I cited in this essay, because psychotherapy traditionally does not engage with the body at all.

I hope that the clarifications of terminology and scope that I have included in this draft will address your concerns about the quality of the writing. Thank you for your support of the topic as being pertinent to specific audiences in the context of contemporary life. 

Round 2

Reviewer 2 Report

Comments and Suggestions for Authors

No additional ccomments